# Early Childhood between a Rock and a Hard Place: Early Childhood Education and Students’ Disruption in Khyber Pakhtunkhwa Province, Pakistan

**DOI:** 10.3390/ijerph19084486

**Published:** 2022-04-08

**Authors:** Jan Alam, Muhammad Azeem Ashraf, Samson Maekele Tsegay, Nadia Shabnam

**Affiliations:** 1Research Institute of Education Science, Hunan University, Changsha 410082, China; janalam.jk@gmail.com; 2School of Education and Social Care, Anglia Ruskin University, Young Street, Cambridge CB1 1PT, UK; samex221@gmail.com; 3Department of Health Profession Education, National University of Medical Sciences, Rawalpindi 46000, Pakistan; nadia.shabnam@numspak.edu.pk

**Keywords:** early childhood education, overcrowded classroom, in-service training, teaching–learning materials, disruptive behaviors, Pakistan

## Abstract

Looking through the lens of ecological system theory, this paper used a mixed-method approach, based on 20 interviews and 208 Early Childhood Education (ECE) teacher questionnaires, to elaborate the position of ECE in Pakistan. The study indicates that ECE is between a rock and a hard place in Khyber Pakhtunkhwa, Pakistan. The findings further show that ECE is provided by less qualified and inexperienced teachers, who give less attention to the physical and psychological needs of the students. The classrooms are overcrowded and lack relevant teaching–learning materials. Moreover, the single-teacher policy and overcrowded classrooms hinder students’ motivation, the delivery of quality education and the development of good behaviors. These challenges are also the main causes of students’ dropouts. This paper increases people’s understanding of ECE and its challenges in Pakistan. For ECE development, the paper recommends separating ECE from primary schools and giving it a budget to purchase adequate and relevant resources.

## 1. Introduction

Early Childhood Education (ECE) in Pakistan is offered to children between three and six years old [1]. It is viewed as the basis for good behaviors and a means to manage social, economic and political challenges as early as possible [2,3]. Effective ECE puts great emphasis on the health and well-being of students [4], which has a lifelong imprint on their characters [5,6]. However, the ECE situation in Pakistan is quite uneasy, where the sector is caught between a rock and a hard place, due to human and material resources [1]. To ease such a problem, the Australian Aid Program inaugurated a project in collaboration with Save the Children to appeal to students in ECE and improve their experiences through teachers’ training [7]. However, this is less likely to occur within a short period in Pakistan, as there is a dearth of ECE-trained teachers [8]. Despite that ECE has earned its status within primary schools in the 1970s [6], it is still in its developmental stage and lacks its own budget to solve the meagre human and material resources [1]. Khan (2018) also argued that there is a lack of political commitment in making ECE a priority area in government planning [3].

Research indicated that budget constraints cause a lack of adequate space, learning resources and teacher’s training [9,10], making the ECE sector less attractive and ineffective [11]. The National Plan of Action (2001–2015) acknowledged the need for specific ECE rooms, teachers, teaching kits and learning materials [12]. In addition, the National Education Policy highlighted the ECE shortcomings that hinder the provision of standardized services [13]. In most of the provinces, the education departments do not meet the ECE needs with regard to the teaching–learning materials. Moreover, the pre-service and in-service training does not match the learning needs of the ECE students, because the task of teaching young children is handled by primary-school teachers who have less training in the interactive and playful methods of teaching young children [6]. This forces many school-age children to be disinterested, demotivated and, finally, out of school, as shown in Figure 1 below.

The delivery of effective ECE and students’ disruption have gained universal attention and become a major issue of access and equity in the developing world. Effective ECE enhances broadmindedness and social integration, leading to peace, development and social harmony [15]. Developed countries focus on rudimentary education to ensure a nonviolent and efficacious social life, but ECE in Pakistan faces human and material resources which are causing low literacy rates and social tribulations [3,6]. This suggests that the ECE sector is caught between a rock and a hard place, showing students’ disruption and the country’s unsatisfactory achievement and low position in education. Different international organizations, such as UNICEF, Australian Aid and Save the Children, are working with Pakistan to expand access to ECE. However, the country still needs to resolve the scarcity of qualified teachers and overcrowded classrooms [6,16]. These challenges are causing students’ disruption and affecting the quality of education at an alarming rate. For instance, in 2018, more than 26% of the ECE students in the province of Khyber Pakhtunkhwa discontinued their education in the first year (see Figure 2 below). Although we could not find national data on ECE to compare, the figure relates to the dropout rate of primary education (29.6%) in Pakistan in the same year [14].

Despite some studies being conducted in ECE in Pakistan, there is still a dearth of research on various aspects of ECE, particularly regarding teachers’ experiences. Pardhan (2012) highlighted the contrast between developmentally appropriate methods of teaching and actual practices in ECE classrooms [17]. Shakeel and Aslam further elaborated that the teaching and learning process in ECE is critical to academic success, holistic development and future achievements [18]. They also emphasized the relevance of teacher training and learning materials. Ghazi et al., also indicated the importance of relevant stationery, charts, projectors, qualified teachers and appropriate teaching methods to engage students in the teaching and learning process [19]. However, there is a scarcity of qualified ECE teachers, learning materials and physical facilities to help students learn via discovery, social interaction and exploration methods [20]. This study, therefore, explored the perspectives of teachers by taking three significant variables and their effects on students’ disruption in classrooms: crowded classrooms, teaching–learning materials and teacher’s in-service training. 

The ECE teachers’ description of ECE as a situation between a rock and a hard place emanates from factors related to teachers’ in-service training, teaching–learning materials and overcrowded classrooms, which also predict students’ disruption in the light of Bronfenbrenner’s Ecological System Theory (EST). This study is based on the following two overarching research questions:What are the teachers’ perspectives about overcrowded classrooms, absence of teaching–learning materials and teachers’ lack of in-service training?How do overcrowded classrooms, absence of teaching–learning materials and teachers’ lack of in-service training influence students’ disruption?

Based on a mixed study, the paper provides a robust explanation of teachers’ perspectives on the factors that affect students’ disruption and dropout. It also contributes to a better understanding of ECE conditions in Pakistan and leads to the production of recommendations informing the delivery of quality ECE services.

## 2. Literature Review

### 2.1. Teacher’s Training and Beliefs

Teachers who have adequate pedagogical and subject-matter knowledge are vital for effective education [14]. They are the chief protagonists that strive for students’ character building, using appropriate teaching techniques and resources [21]. However, Pakistan lacks such teachers, and this lack is directly attributed to the poor quality of education in the country [22]. The low level of teachers’ knowledge and skills affected the instructional methods and delivery of lessons, causing students’ low motivation and high dropouts [23]. This suggests that many teachers are unaware of the students’ needs and interests [24,25], and this unawareness emanates from lack of teacher training and facilitation [26]. Lack of teacher training is associated with non-interactive pedagogy [27] and low academic performance [28]. Such a situation at least requires a strong belief and commitment of teachers that serves as a bridge between the school realities and teachers’ training [22]. It can strengthen the teachers’ desire to improve their capacity and stimulate students’ motivation to learn [29].

Teachers’ beliefs are metaphorically a contextual filter to screen classroom experiences. In most cases, teachers’ training and qualifications significantly influence the students’ beliefs, understanding, needs and interests [30]. Furthermore, research indicates that teachers’ beliefs nourish their thinking, decision-making and classroom practices [31], which are essential for effective learning and interaction [32,33], and following school regulations, including discipline [34]. This suggests that students’ learning is greatly supported or influenced by teachers’ beliefs, because it encourages the development of various skills and attitudes, which are vital for school readiness and achievement [35]. Moreover, teachers’ beliefs contribute to the achievement or failure of students’ socialization through implicit or explicit developmentally appropriate practices and theories [36]; and this strengthens teacher–student relations and promotes positive behaviors and class interactions [37]. In addition, a teacher’s belief also paves the way for planning, goal setting and students’ motivation [32,38]. Overall, a teacher’s belief is a driving force for understanding students’ needs and interests and delivering effective ECE with the available resources [39].

### 2.2. Learning Materials and Space

In addition to teachers’ in-service training and facilitation, teaching materials are the basic components for effective instructional delivery and students’ socio-emotional development [40]. As indicated above, pedagogical capabilities are the cornerstones of ECE, associating children’s learning with their emotions, communities and the broader society [41]. Similarly, adequate and relevant teaching materials are vital instruments for solving problems and connecting theory with practice [42]. This helps shape pedagogy based on students’ needs and interests [42] and helps minimize students’ disruption [43].

Moreover, the availability of enough learning space and places, such as classrooms, has a significant effect on students’ learning [3,44]. However, despite its high importance [45], the limitation of space is a critical issue in public schools [46], and this hinders the efforts made to meet the physical needs of the students in Pakistan [8]. Furthermore, an overcrowded classroom triggers academic and behavioral challenges [47], which interfere with the pedagogical practices [48] and cause low teacher–student interactions [49]. Large class size also affects students’ learning, safety and well-being. It hinders the use of a constructivist approach of teaching [17,50] and clean and well-conditioned classrooms [51]. Such a situation hinders effective education [46] and leads to students’ disruptions [52], poor performance and high dropouts in Pakistan [46,53].

### 2.3. Theoretical Framework: Bronfenbrenner’s Ecological System Theory (EST)

Watson and Rayner [54], Skinner [55] and Bandura and Walters [56] studied the role of the environment in the growth and nourishment of a child and came up with various ideas. For example, Watson and Rayner [54] considered children to be passive beings, similar to clay, whereas Skinner [55] posited that learning is a response of an organism to the environment. Bandura and Walters [56] also stated the importance of observation and imitation in learning. Then Bronfenbrenner combined all of these theories, while noting that learning and behavior are shaped by the system of interrelationship in the environment [57]. This makes the Ecological System Theory (EST) valuable, dominating other theories related to child development and becoming a beacon for researchers across disciplines, particularly in the fields of education, psychology and sociology [58,59]. According to Bronfenbrenner, the following is true:


*“The ecology of human development is the scientific study of progressive, mutual accommodation throughout the life course between an active being and changing properties of immediate setting in which the developing person lives. This process is affected by the relations between these settings and by the larger contexts in which the settings are embedded”.*
[60]

Ecology is the scientific study of organisms and how they interact with organic or inorganic surroundings around them [61]. EST covers information happening in all four systems, namely micro, meso, exo and macro, around a child [62]. EST also indicates the role of a child’s interaction in learning new skills, knowledge and concepts [58,63]; and the advantage of problem-solving skills to acquire more knowledge in the company of other participants, mainly teachers, peers and parents [64].

Konza [65] reflected that the classroom, in the light of Bronfenbrenner’s EST, could help children connect with their communities and the wider society. This could also create an arrangement for children with a teacher, as an interactive component, to have principles for system endurance and classroom facilitation. Research has shown that interactive pedagogy supports students’ stability and ability to determine organizational patterns for communication and behavioral expectations [50,66]. Moreover, Allen (2010) noted that students’ understanding and behaviors according to EST are lenient to class size and teaching practice [67]. Classroom ecology consists of students, teachers, learning materials and various activities that directly influence behaviors and shape disruptions that cause conflict between matter and manner [68]. Similarly, Piaget and Vygotsky claimed that teachers’ instruction is significant for students’ construction of learning in radical and social background [14]. This further affects the development of cognition, language, motor and adaptive skills associated with socio-emotional operations [69], students’ learning motivation, good relationship and problem-solving skills [70].

According to Bronfenbrenner’s EST, a child has two essential binary layers, the upper and the supportive layers [71]. The upper layer consists of the immediate settings, such as the home, school, street and playground, where the child’s life and behavior are stimulated. The supportive layer includes the physical and geographical settings and the context of the institutions around the child [72]. Ettekal and Mahoney further explained that there are three fundamental components of EST: person, process and context. Contemporary theories suggest that human development happens due to human interaction within a specific context or environment [73]. Bronfenbrenner’s EST also highlights that teachers’ class preparation relates to four different systems: microsystem, mesosystem, exosystem and macrosystem [58,62,74]. The microsystem explains the individual and the teachers’ collective potential, the mesosystem discusses the relationship of various contexts relevant to teachers, the exosystem highlights organizational information and the macrosystem postulates information regarding the influence of society and school legislation. All of these aspects directly or indirectly affect teachers’ training and classroom situation and, thus, influence students’ disruption.

## 3. Methodology

Creswell argued that pragmatism allows researchers to use multiple methods and forms of data collection and analysis to better understand the problem under study [75]. Therefore, a convergent mixed-method design was adopted to ensure a parallel data-collection process and conduct a separate analysis for triangulation and validation purposes [76]. The study was conducted in Peshawar District, located in the Khyber Pakhtunkhwa (KP) province of Pakistan, for two main reasons: proximity to the researchers for data collection and significant ECE-student dropout rate in the province (see Figure 2 above). Figure 3 shows the geographical location of Pakistan and the the Khyber Pakhtunkhwa (KP) province.

This study employed a mixed approach consisting of two main phases: questionnaire and interviews. We now discuss the two phases and the research ethics in detail.

### 3.1. Questionnaire

The total number of primary-school teachers in Peshawar district was 1033, out of which 20% were selected through a simple random sampling technique for the quantitative phase of the study. In total, 208 participants (104 boys and 104 girls) were selected for the quantitative part of the study. The teacher participants had a good English capacity, and the language was used for the survey.

The demographic information highlighted in Table 1 indicates teachers’ qualifications and ECE training. In Pakistan, ECE is part of primary school. Hence, teachers are required to have a primary teaching certificate (PTC), which is relevant for primary-level teaching, but not for ECE. However, as shown in the table below, 182 of the entireparticipants have no ECE certificates, although they had taught for many years.

In terms of age, 132 of the total participants were above 45 years old. In addition, 58 out of the 208 teachers had higher secondary-school qualifications, whereas the rest had at least a graduate-level qualification.

This is a descriptive study aimed to describe the current situation of ECE as a predictor of students’ disruptions. An online survey was used to collect data from the ECE teachers. A twenty-two items Likert Scale questionnaire, with five scales, from “Not at All” to “Very Good”, was applied for data collection. The questionnaire was assembled by merging two adapted questionnaires used by Ntumi [77] and Ghazi [78]. The data were subjected to statistical treatment by using the Partial Least Squares Structural Equation Model (PLS-SEM) via SmartPLS 3.0, which is significant for analyzing the key predictors of outcome variables.

### 3.2. Interview

This study applied semi-structured interviews with 20 ECE teachers to obtain adequate and relevant information by asking additional questions based on the interviewee’s response. Out of the ten circle schools of Peshawar district, purposive sampling was used to select one male and one female ECE teacher per school to ensure school and gender representations (see Table 2). Purposive sampling is a non-probability sampling technique that is used to focus on particular characteristics of a population based on the study’s objectives and the researcher’s knowledge of the population under study [75]. Contrary to the survey, the interviews were conducted in Urdu to allow better communication and expression of ideas and, then translated to English. Since two of the researchers are fluent in Urdu and English, the backtranslation of the interview questions was smooth and effective [79]. Each interview lasted between 45 and 60 min. The interview questions, developed from the research questions, covered topics about teachers’ perspectives on in-service training, teaching resources and class size in relation to students’ disruptions.

In order to ensure the trustworthiness of the qualitative data, different strategies were adopted based on the guidelines provided by Shenton [80] and Nowell et al. [81]. First, the credibility of the qualitative data was addressed by the researchers who were born and have been working in the region [80,81]. Two (J.A. and M.A.A.) of the researchers are familiar with the Pakistani context. Therefore, their positionality, including their personal experiences, beliefs and linguistic capabilities, was significant to get access to the participants and collect relevant data [80,82]. The researchers were also able to make personal reflections of the interviews to assess the impact of their positionality on the interview process and outcome [80,82]. Then the transferability of the data was carried out by other researcher (S.M.T and N.S.) to check the external validity or applicability of the findings to other situations [80,82]. In the next phase, all authors ensured the dependability and reliability of the data by adopting a relevant research design and implementing it in fieldwork. We also believe that the study adopted proper research methods to collect and analyze the qualitative data [80,83]. In particular, a thematic data analysis was used to select significant features from the large quantity of data produced from the interviews [81,84]. Finally, the confirmability of the data was checked, ensuring that the findings of the study are presented in the results as participants’ ideas and experiences, instead of the characteristics and preferences of the researchers. Thus, these different measures ensured the credibility of the data in general and the qualitative data in particular.

A thematic analysis was used to analyze the data collected through semi-structured interviews in order to summarize the key features and offer a rich interpretation of the data [81]. The interview data were arranged into meaningful units for the general structure description [85]. The data analysis started with descriptive coding via brief comments, and then descriptive cluster coding was applied for relating the data to the research questions. This was followed by deriving themes from the data and analyzing them in relation to the quantitative results and the literature review. Finally, we analyzed the qualitative and quantitative data and triangulated the findings to explore the divergence or convergence [86]. For triangulation purposes, the data were discussed by integrating both data types through a narrative form [87], and complementing [88] and supporting each other [89].

### 3.3. Research Ethics

We followed appropriate ethical considerations throughout the research process (see National Commission for the Protection of Human Subjects of Biomedical and Behavioral Research, 1978) [90]. The Hunan University Ethics Committee approved the research. In addition, permission to conduct the study was obtained from the Directorate of Education in KPK, Pakistan. All of the participants were informed about the study’s objectives and their right to withdraw from the study at any time. We used the consent form to provide adequate information and ensure their willing participation in the study. Moreover, the names of the participants were changed with pseudonyms to protect the privacy and anonymity of the participants [91].

## 4. Findings

### 4.1. Quantitative Data Results and Analysis

A Partial Least Squares Structural Equation Model (PLS-SEM) via SmartPLS 3.0 was applied to analyze the quantitative data. We performed a measurement model (internal consistency reliability, discriminant validity and convergent validity), structural model (R square and path coefficient) and multigroup analysis.

The internal consistency reliability and convergent validity are summarized in Table 3. All the constructs and indicators were founded to meet the reflective measurement criteria. The indicators’ reliability is achieved as the outer loadings (λ) are (F-Loading > 0.832) at the minimum. The average (AVE) values are (AVE > 0.769) to the minimum, which indicates that convergent validity is achieved. The Composite Reliability (CR) is (CR > 0.945), which is above 0.70 and required value to secure internal consistency. The rho-A indicates reliability indices for each construct, which is (rho-A > 0.928). The Variance Inflation Factor (VIF) is used to highlight the collinearity of the items. The results from the test highlighted that the measurement criteria were achieved.

Discriminant validity makes a construct unique compared to its counterparts [92] and determines the discriminant validity that the authors proposed. The values must be greater than all others in the rows and columns [92]. As Table 4 highlights, the diagonal values are greater than all others, confirming the discriminant validity.

The structural model indicates the R Square, effect size, t-value and *p*-value. Before the structural model, the collinearity of the construct was tested in Table 4, using the Variance Inflation Factor (VIF), which is less than threshold 5 [92], and with no collinearity issue. A positive significant influence of teachers’ training on students’ disruptive behaviors is indicated via statistical values with (β = 0.326, t-value = 3.631 and *p* = 0.000, *p* > 0.001). A crowded classroom also indicates significant influence on students’ disruptive behaviors via statistical values (β = 0.360, t-value = 4.445, *p* = 0.000 and *p* > 0.001). However, the third factor, learning materials, has no significant influence on students’ disruptive behaviors in light of the statistics in Table 5.

### 4.2. Interview Findings

#### 4.2.1. Teachers’ Perspectives on Students’ Disruption

The respondents indicated that students’ disruption in early classrooms is expected because the students are mostly caught between a rock and a hard place. Most of the teachers are not qualified to teach ECE, which affects their awareness about the physical and psychological needs of the students. Moreover, the classrooms are overcrowded and lack appropriate learning materials. These challenges make Peshawar District no different from the other districts of Pakistan, which experience a lack of resources and considerable out-of-school children [5,93].


*“Disruption is the dominant character of young children because the school environment is not conducive to their growth and development. The fact is that teachers for ECE are least qualified, which hinders them from knowing the needs of young children. Moreover, their materials are old-fashioned and not according to the syllabus”.*
(Johar)

Without having a conducive environment, most actions of young children might lead to disruption. Therefore, teachers’ practices and learning materials are essential components of a classroom environment that engage students in teaching and learning and could contribute to controlling disruption [94].


*“Students in early classrooms [ECE] are very young and require more attention. They are less than five years of age and often need social and behavioral training. If they are not fully attended to in classrooms, they snatch items from one another and disturb the entire class. In most cases, such cases are caused by negligence from teachers and the school authority”.*
(Asma)

As can be seen above, Asma states that teachers and school authorities are responsible for students’ social learning and discipline. This makes teachers shoulder double responsibilities in shaping their students’ behavior and motivation to learn.

#### 4.2.2. Factors That Affect Students’ Disruption

This section discusses the impact of in-service training, class size and teaching aids on students’ disruption. In line with the quantitative results and many other studies, the interview data indicate that a lack of teachers’ in-service training and overcrowded classrooms significantly influence students’ disruptive behaviors [37]. Moreover, contrary to the quantitative results, the findings suggest that teaching and learning materials indirectly influence students’ disruptions.

The entire group of participants indicated the importance of ECE teachers’ in-service training to create a positive learning environment and engage students in class activities.


*“Controlling students in a classroom is not the aim of teaching. Instead, teaching or education should create an environment for their guidance and lead them to independent learning and character-building, which is only possible via effective teacher training ”.*
(Asif)

Asif’s statement was supported by most of the participants. They further argued that most of them had not received any training related to ECE, because ECE is a part of primary schools in Pakistan. In most cases, the aged and least qualified teachers are assigned to teach ECE. The training offered in collaboration with other institutions is often short-lived. For example, Save the Children and Provincial Institute for Teachers’ Education (PITE) started to offer workshops and training for ECE teachers, but these were discontinued after a very short period, due to budget problems. In addition, the teachers who participated in the ECE workshops and training could not practice the skills they acquired in their classrooms, due to a lack of teaching resources or interference from the school authority. In some cases, the teachers were not assigned to teach ECE, as stated by Asad below:


*“I have a personal interest in ECE and, for this very purpose, I enrolled myself in a private institute to get a certificate. Even though I was eligible to teach young children, I have not been allowed by the school authority since they consider ECE the easiest job. Only aged teachers are permitted to teach them due to their age and minimal qualification. However, many of those teachers have been requesting help or replacements from teachers with ECE qualifications”.*
(Asad)

Asad’s statement indicates that the school authorities consider ECE students the easiest to teach, even without relevant training. However, the participants declared the importance of in-service training, which gives teachers insight into the students’ needs and interests to create a learning environment that motivates students to learn rather than behave disruptively. Students run from one corner of the class to the other, shouting, quarrelling and abusing each other, and they even discontinue their education when the teachers fail to understand their needs and interests. For example, one of the participants indicated the following:


*“Teaching young children is a difficult job that requires effective in-service training. Because one has to attract the students and motivate and engage them in activities to reduce disruption”.*
(Asma)

Asma’s excerpt suggests that teachers need adequate training that fits the level of students they teach to support the teaching and learning activities effectively and minimize the students’ distruption and dropout. However, this study also shows that class size affects the delivery of lessons and interaction of students in a class.

Habib (2015) indicated that, in Khyber Pakhtunkhwa, Pakistan, the standard class size is 40 students per class [95]. Nevertheless, the participants of this study noted that there are 50 to 80 children in ECE classrooms. Moreover, there is a single-teacher policy, which makes it difficult for the less qualified and aged teachers to focus on all the children in the class. It creates a noisy environment where many students often leave the class or move within it, playing or fighting with peers. Asad and Salma stated the following:


*“In an overcrowded classroom, it is difficult for a single teacher to check the students’ progress, motivation, and discipline. The teacher at least needs material support and a teaching assistant to control or minimize students’ disruption”.*
(Asad)


*“As the need of young children for space and place is not being fulfilled in my classroom, most of the students are not interested in coming to the congested class. Many do not come to school regularly and others discontinue their education”.*
(Salma)

Asad’s quotation is central to the need for an ideal class size that teachers can manage and in which students engage without a problem. The congestion of many students in a class is ineffective for young children’s behavioral and academic development and pushes students to interrupt their study, as Salma highlighted.

To make things worse, the ECE lacks adequate and relevant teaching materials, and the teachers follow traditional teaching methodology in their classrooms. For example, some schools only possess color charts, maps and boxes containing numbers and letters, while others have additional resources, such as toys. The government also provides books. However, the provided resources are not enough, and many parents cannot afford notebooks, pencils, colors, crayons and other relevant learning materials, due to socioeconomic problems. Asim explained the situation as follows:


*“There is a lack of adequate teaching materials to help my students learn better. I was provided with a little fund by the school to buy some low-cost teaching materials. Nevertheless, the money was insufficient, preventing me from purchasing anything relevant for my class”.*
(Asim)

As most parents cannot afford to buy their children the required materials, some teachers try their best to buy low-cost materials from the money they get from the school and at their own expense.


*“The materials I use in my classroom are old-fashioned and do not match the syllabus and students’ needs. So, I buy low-cost materials such as a clock, ball, balloon, and colors at my expense”.*
(Salma)

Contrary to Salma’s idea, some participants argued that it is not easy for teachers to buy teaching materials on their own. Instead, they should become somehow innovative and facilitate the students’ activities and control disruption by using different methods. Regarding this, Johar highlighted the following:


*“I have not been provided with a single material to help me improve the interest and motivation of my students. To cover the loopholes, I use little pebbles and draw circles on the ground to engage students in different activities and help them better understand. In so doing, I can control their disruption and dropouts”.*
(Johar)

Overall, the findings suggest that teaching and learning materials indirectly affect students’ disruption. They affect the teaching and learning process and students’ motivation to learn, which impacts students’ satisfaction and disruption.

## 5. Discussion

The questionnaire and interview data suggested that a lack of teachers’ in-service training, overcrowded classrooms, and teaching and learning materials stimulate students’ disruption, which puts the situation of ECE between a rock and a hard place. Research shows that ECE has passed through developmental stages since 1947 but has brought little change, due to the challenges mentioned above [93]. This indicates that the ECE sector needs substantial work on providing teachers’ in-service training, supplying relevant teaching–learning materials and lowering the class size to offer quality education and tackle students’ dropout. In this line, the Ecological System Theory states the impact of the teaching context on students’ learning and behaviors [74]. This study also shows that a lack of conducive educational settings leads to disruption [96], as clearly demonstrated in the measurement model in Table 3 above.

Demographic information in Table 1 and Table 2 highlighted the challenging situation of ECE, due to the provision of unqualified and aged teachers that suggests that ECE was not considered as an essential stage in students’ development. First of all, there was no special in-service training and, if there was any, it did not continue for long, due to a lack of budget. Thus, most ECE teachers did not go for training or workshop, as they did not find a chance or thought nothing would change. As some of the participants indicated, teachers are not assigned to ECE based on their qualifications, because the school authorities thought that ECE was the easiest to teach. Some particpants also noted that they were not able to use any new skills they learned, due to lack of resources and overcrowded classrooms. The statistical results of the structural model in Table 5 illustrated the significant effect of teachers’ in-service training on students’ disruption with (*p* < 0.001). The National Educational Policy also states the need for at least two years of compulsory training for ECE teachers [13], but it has not been practical. According to [93], the ECE training often lasted two weeks and was later discontinued due to lack of budget. Therefore, there is a need for collaboration between ECE and higher education institutions to offer relevant and timely training on ECE classrooms and teaching methods.

As highlighted in the measurement model in Table 3, ECE classrooms are quite congested, overcrowded and based on a single-teacher policy. This causes students’ boredom and disruptive behavor. Table 5 further highlighted the significant influence of overcrowded classrooms on students’ disruption because (*p* < 0.001). The study elaborates the National Plan of Action 2001–2015 and suggests that the action plan should include an ideal class size, teachers’ training and learning materials for ECE [12]. ECE in the Peshawar District is part of primary school, where it is placed in a specific classroom of a school, regardless of the number of ECE students registered for the school in a particular year. The availability of a single teacher for a class can also create boredom among students. Moreover, it is difficult for a single teacher to manage large classes and assess the progress of every student. This often decreases students’ motivation to learn and leads to disruption and dropout. On the other hand, in overcrowded classrooms, the teachers face a scarcity of time and teaching and learning materials to engage students and improve their learning [50]. Engaging and motivating students also improve their discipline, because behavioral development is influenced by classroom environment and practices [97,98].

Table 3 indicated the challenges regarding the lack of teaching and learning materials, but Table 5 highlighted that they have no significant effect on students’ disruption (*p* > 0.05). The reason is that most of the teachers are providing low-cost materials from the budget they received from the school and at their own expense to fill the gap. According to Bronfenbrenner’s Ecological System Theory, space, place and available resources are the immediate settings around young children [99], and they create the learning environment [100]. However, the qualitative data showed that enough resources were not provided, due to lack of budget. This suggests the need to separate ECE from primary schools to secure their own funding for buying essential resources. Furthermore, the departure of ECE from primary schools can enable ECE to have more than one teacher and classroom where they can place children according to their age and needs as per the National Educational Policy.

## 6. Conclusion and Implications

The study depicted ECE in Khyber Pakhtunkhwa, Pakistan, as a situation caught between a rock and a hard place. On one side, it is run by unqualified teachers who lack adequate awareness of young children’s physical and psychological needs. On the other side, the classrooms are overcrowded and lack proper teaching and learning materials. These challenges affect the young children’s learning environment and, thus, hinder their behavioral development and learning outcomes. Overall, the inclusion of ECE in the primary section has affected the provision of relevant resources and trained teachers. ECE has no adequate budget for such resources because more attention has been given to primary school classes. Moreover, the ECE students are placed into a single class, regardless of their number. This caused students’ disruption, absence and even the interruption of their education. The findings imply that ECE in Pakistan has to be separated from primary schools in order to deliver quality services, particularly to students who cannot afford to buy teaching resources. The split from primary schools can allow ECE to have its own budget, more classrooms, energetic and trained teachers and the attention of educational authorities. If this is not performed, then the country might continue facing a low literacy rate, a large number of dropouts and, thus, socioeconomic disparity.

This article provides a new perspective concerning ECE in Pakistan. It offers a better understanding of the challenges faced by the ECE sector and their impact on studnets’ disruption and dropouts. However, the research article is not without its limitations. It was conducted in one district of the Khyber Pakhtunkhwa, Pakistan. In addition, this article was based on data collected from teachers. Therefore, similar research from other provinces of Pakistan could expand the findings and allow for comparative analysis. Moreover, studies that capture the view of other stakeholders, including parents and school authorities, are necessary to develop a more robust understanding of the topic.

## Figures and Tables

**Figure 1 ijerph-19-04486-f001:**
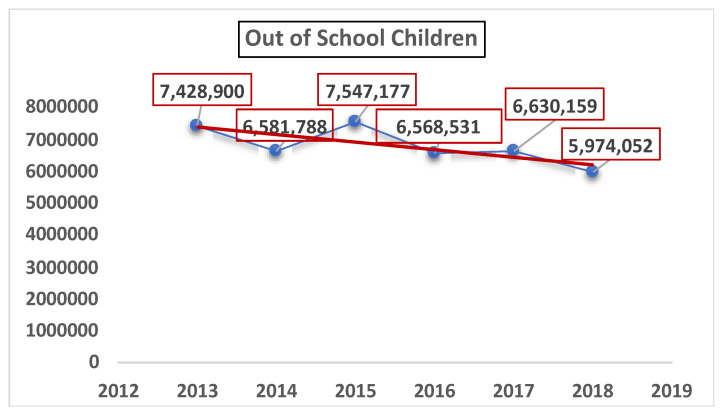
Out-of-school children in Pakistan [14].

**Figure 2 ijerph-19-04486-f002:**
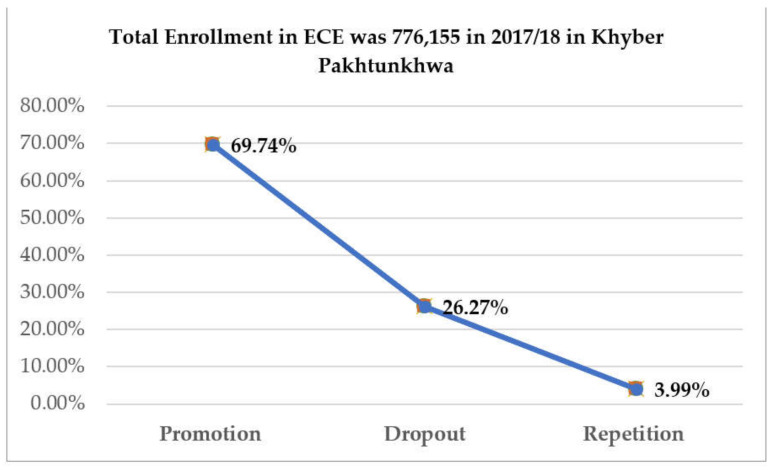
ECE Enrolment in the Province of Khyber Pakhtunkhwa [14].

**Figure 3 ijerph-19-04486-f003:**
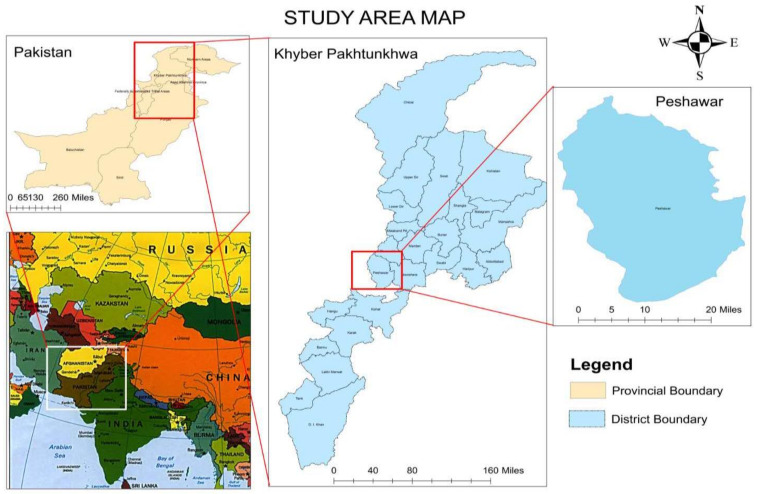
GIS map for the study site.

**Table 1 ijerph-19-04486-t001:** Teachers’ demographic information.

Name	Category	Frequency (%)
Teacher Gender	Male	104 (50.0)
	Female	104 (50.0)
Age	Below 30 Years	6 (2.9)
	31 to 35 Years	12 (5.8)
	36 to 40 Years	20 (9.6)
	41 to 45 Years	38 (18.3)
	46 to 50 Years	58 (27.9)
	Above 50 Years	74 (35.6)
Qualification	Intermediate + PTC	58 (27.9)
	Under Graduate + PTC	88 (42.3)
	Masters + PTC	50 (24.0)
	M.Phil + PTC	11 (5.3)
	Doctoral	1 (0.5)
ECE Certificate	Yes	18 (8.7)
	No	182 (87.5)
	In Progress	8 (3.8)
Experience	0–5 Years	13 (6.3)
	6–10 Years	16 (7.7)
	11–15 Years	29 (13.9)
	16–20 Years	33 (15.9)
	21–25 Years	48 (23.1)
	Above 25 Years	69 (33.2)

**Table 2 ijerph-19-04486-t002:** Qualitative phase teachers’ demographic information.

Pseudonym	Gender	Age	Rural/Urban	ECE Certificate	Qualification	Experience/Years
Asma	Female	45	Urban	No	Master	19
Abbas	Male	49	Urban	No	Bachelor + PTC	27
Aisha	Female	35	Rural	No	Master	9
Asad	Male	58	Rural	No	Intermediate + PTC	30
Asif	Male	44	Urban	No	Master	19
Sadaf	Female	29	Urban	Yes	Master	6
Salma	Female	51	Rural	No	Intermediate + PTC	27
Sana	Female	55	Rural	No	Bachelor + PTC	28
Asim	Male	41	Urban	No	Master	16
Mahjabeen	Female	56	Urban	No	Bachelor + PTC	25
Ameen	Male	57	Rural	No	Bachelor + PTC	29
Johar	Male	58	Rural	No	Intermediate + PTC	28
Kalsoom	Female	46	Urban	No	Bachelor + PTC	22
Waseem	Male	38	Urban	Yes	Master	7
Mehmood	Male	53	Rural	No	Bachelor + PTC	26
Noreen	Female	47	Rural	No	Bachelor + PTC	19
Ahmad	Male	59	Urban	No	Intermediate + PTC	27
Khalida	Female	53	Urban	No	Intermediate + PTC	25
Kashif	Male	28	Rural	Yes	Master	4
Safna	Female	30	Rural	No	Master	3

**Table 3 ijerph-19-04486-t003:** Measurement model.

	F-Loadings	Cronbach’s Alpha	rho A	CR	AVE	VIF
Disruptive Behaviors		0.943	0.946	0.959	0.855	
Following Instruction	0.932					4.653
Social Behaviors	0.900					3.247
Discipline	0.934					4.471
Peers Relation	0.933					4.327
In-Service Training		0.940	0.951	0.952	0.769	1.960
Understanding Curriculum	0.917					4.482
Effective ECE Training	0.908					4.158
Understanding Child’s Needs	0.859					3.048
Authority Interest in Pedagogy	0.847					2.896
Check on Pedagogy	0.862					2.803
Experience to Engage	0.867					2.931
Overcrowded Classroom		0.927	0.928	0.945	0.775	2.099
Individual Attention	0.895					3.511
Motivation	0.892					3.807
Activities	0.886					3.050
Space for Each Student	0.896					3.582
Assessment	0.832					2.230
Learning Materials		0.941	0.942	0.958	0.850	1.750
Available Materials	0.919					3.632
Materials in Students’ Access	0.922					4.057
Activities via Materials	0.923					4.057
Support for Materials	0.923					3.813

**Table 4 ijerph-19-04486-t004:** Fornell–Larcker criterion.

	Disruptive Behaviors	Teachers’ Training	Crowded Classroom	Learning Materials
Disruptive Behaviors	**0.925**			
Teachers’ Training	0.375	**0.877**		
Crowded Classroom	0.465	0.668	**0.881**	
Learning Materials	0.371	0.579	0.614	**0.922**

**Table 5 ijerph-19-04486-t005:** Structural model results.

Relationships	Std. Beta	Std. Error	t-Values	*p*-Values	CI LL	CI UL	Decision
Teachers’ Training -> Disruptive Behaviors	0.326	0.090	3.631	0.000	0.144	0.494	**Supported**
Crowded Classroom -> Disruptive Behaviors	0.360	0.081	4.445	0.000	0.207	0.517	**Supported**
Learning Materials -> Disruptive Behaviors	0.148	0.077	1.853	0.064	−0.017	0.300	Not Supported

## Data Availability

The dataset for this study can be asked from corresponding author (azeem@hnu.edu.cn) on reasonable request.

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
