# Peer review of "Early Childhood between a Rock and a Hard Place: Early Childhood Education and Students’ Disruption in Khyber Pakhtunkhwa Province, Pakistan"

_ijerph, 2022, doi:10.3390/ijerph19084486_

Round 1

Reviewer 1 Report

The topic and the research work are interesting. The methodological approach, considering the nature of the research, is right. Using questionnaires and interviews seem to be a correct way to get data about the research topic.

Comments and suggestions

Sentences that can be reworded for a better understanding

Lines 143 & 144.- ‘It hinders the use of the constructivist approach of teachings (Pardhan, 2012) and hygienically well-conditioned classrooms (Dean, 2005) and hinders teachers from using‘.

Format issues

  • Line 38.- In addition, the Nationtional Education Policy 2017, p. 27) à National instead of Nationtional; besides, there is a single closing parenthesis (without the open parenthesis).
  • Line 48 (Figure 1); Line 65 (Figure 2); Line 211 (Figure 3) -> It should be clearer to read if “,” were included in the numbers in Figure 1 & Figure 2. Besides, the fonts used in Figure 1, Figure 2 and Figure 3 are different for the one used in text.
  • Line 66; Line 111; Line 156; Line 201; Line 217; Line 381; Line 453.- There is an extra space in each line that should be removed.
  • Line 84. At the end of the sentence ‘:’ should be used instead of ‘.’.
  • Line 186.- A extra space in needed.
  • Line 196; Line 208.- An odd number of parentheses.
  • Line 224.- A line between each ‘Name’ would make it cleared to read Table 1.
  • Line 440 -> Instead of demonstratedin’ -> demonstrated in (an extra space in required).

Introduction

Lines 61-63.- The figures showed in Figure 2 are the ones associated to the province of Khyber Pakhtunkhwa and are presented as an example… Are those numbers out layers compared the ones of the other provinces? Are those numbers similar to the other provinces? Are those number homogeneous in Pakistan?

Methodology

Lines 218 & 219 -> Why did you choose the Khyber Pakhtunkhwa province of Pakistan instead of any other province? Is this province representative? If this latter question is the case, it should be great to include data/research studies/reports to support this assumption. Depending on the answer, lines 90-92 should be re-written.

Findings

Lines 311 & 312.- ‘These challenges make Peshawar District not different from the other districts of Pakistan concerning ECE’. --> Are all districts ‘equal’ in terms of ECE? Are there any research works/data/reports that support the assumption?

  1. Discussion & Conclusion and   6.Implications and Recommendations

You could consider renaming this section to Discussion. It should include the discussion and the implications and recommendations (in a different paragraph or as a subsection within this section).

Conclusions could be inserted in a ‘Conclusion’ section, the last one of the research article (i.e. 6, if the structure of the research work is maintained).

Inconsistencies in the bibliography:

  • Line 538 -> Century;, 69, 22-28.
  • Line 609.- Nationtional Education Policy 2017 à National

Author Response

Reviewers' Comments to Authors:

Reviewer 1

The topic and the research work are interesting. The methodological approach, considering the nature of the research, is right. Using questionnaires and interviews seem to be a correct way to get data about the research topic.

Response: Thank you.

Comments and suggestions

Sentences that can be reworded for a better understanding

Lines 143 & 144.- ‘It hinders the use of the constructivist approach of teachings (Pardhan, 2012) and hygienically well-conditioned classrooms (Dean, 2005) and hinders teachers from using‘.

Response: we have corrected it in the revised article.

Format issues

  • Line 38.- In addition, the Nationtional Education Policy 2017, p. 27) à National instead of Nationtional; besides, there is a single closing parenthesis (without the open parenthesis).
  • Line 48 (Figure 1); Line 65 (Figure 2); Line 211 (Figure 3) -> It should be clearer to read if “,” were included in the numbers in Figure 1 & Figure 2. Besides, the fonts used in Figure 1, Figure 2 and Figure 3 are different for the one used in text.
  • Line 66; Line 111; Line 156; Line 201; Line 217; Line 381; Line 453.- There is an extra space in each line that should be removed.
  • Line 84. At the end of the sentence ‘:’ should be used instead of ‘.’.
  • Line 186.- A extra space in needed.
  • Line 196; Line 208.- An odd number of parentheses.
  • Line 224.- A line between each ‘Name’ would make it cleared to read Table 1.
  • Line 440 -> Instead of ‘demonstratedin’ -> demonstrated in (an extra space in required).

Response: We have corrected these in the revised version. We appreciate reviewers’ suggestions.

Introduction

Lines 61-63.- The figures showed in Figure 2 are the ones associated to the province of Khyber Pakhtunkhwa and are presented as an example… Are those numbers out layers compared the ones of the other provinces? Are those numbers similar to the other provinces? Are those number homogeneous in Pakistan?

Response: We have tried to add this. However, there is a lack of data showing the ECE dropout rate of Pakistan. Most of the data start from primary school.

 Methodology

Lines 218 & 219 -> Why did you choose the Khyber Pakhtunkhwa province of Pakistan instead of any other province? Is this province representative? If this latter question is the case, it should be great to include data/research studies/reports to support this assumption. Depending on the answer, lines 90-92 should be re-written.

Response: We have added a sentence to address the above comment: “The study was conducted in Peshawar District, located in the Khyber Pakhtunkhwa (KP) province of Pakistan, for two main reasons: proximity to the researchers for data collection and significant ECE students dropout in the province (see Figure 2).”

Overall, Khyber Pakhtunkhwa province was not supposed to represent all districts. We could not find enough data showing the ECE dropout rate of all regions in Pakistan to compare and ensure that. Almost all of the data start from primary schools.

Findings

Lines 311 & 312.- ‘These challenges make Peshawar District not different from the other districts of Pakistan concerning ECE’. --> Are all districts ‘equal’ in terms of ECE? Are there any research works/data/reports that support the assumption?

Response: ECE in Pakistan (across all its districts) is characterized by a lack of resources and significant out of school children. We have added a reference to this argument.

  1. Discussion & Conclusion and   6.Implications and Recommendations

You could consider renaming this section to Discussion. It should include the discussion and the implications and recommendations (in a different paragraph or as a subsection within this section).

Conclusions could be inserted in a ‘Conclusion’ section, the last one of the research article (i.e. 6, if the structure of the research work is maintained).

Response:Thank you for this. We have made some arrangements as suggested.

Inconsistencies in the bibliography:

  • Line 538 -> Century;, 69, 22-28.
  • Line 609.- Nationtional Education Policy 2017 à National

Response: We have corrected the above errors and rechecked the entire reference list.

Reviewer 2 Report

1.Authors SHOULD add more related recent studies for the Introduction and Literature review.

2. Authors need to inform the process instrumentation, such as back translation (see, Behr 2017), or content or face validity processes as an initial step establishing the survey statement. 

3. The trustworthiness of the qualitative data should be elaborated 

4. The questions over how the interview questions were developed are not explained in the method sections.

5. Authors are required to open their instrument as well as interview question data; important for other researchers to adopt or adapt

6. The limitations of the study and recommendations for further studies are not reported; please elaborate

Author Response

1.Authors SHOULD add more related recent studies for the Introduction and Literature review.

Response: We believe that we used most of the recent ECE articles regarding Pakistan, as there are not many. We have also added some recent studies.

  1. Authors need to inform the process instrumentation, such as back translation (see, Behr 2017), or content or face validity processes as an initial step establishing the survey statement. 

Response: English was used for the survey since the participants were teachers who can speak English (which is also the official language of Pakistan along with Urdu). However, Urdu was used for the interview. And few sentences have been added to elaborate on this.

  1. The trustworthiness of the qualitative data should be elaborated 

Response: We have 20 participants which are considered enough number for a qualitative study. We have also further elaborated the method, which can contribute to the trustworthiness of the qualitative data. However, it is important to note that the data are personal stories of the participants, which should not be put to judgment.

  1. The questions over how the interview questions were developed are not explained in the method sections.

Response:  We have added a sentence to clarify this.

  1. Authors are required to open their instrument as well as interview question data; important for other researchers to adopt or adapt

Response: We have explained the instrument which is assembled by merging two adapted questionnaires used by Ntumi (2016) and Ghazi et al. (2013). However, we did not see the value of opening our interview questions as researchers are less likely to repeat our interview questions into their study. However, we will consider uploading it to the journal’s system if it is found to be of high value.

  1. The limitations of the study and recommendations for further studies are not reported; please elaborate

Response:  We have included limitations of the study and recommendations for further studies to the conclusion.